# Self-organization of songbird neural sequences during social isolation

**Emily L Mackevicius, Shijie Gu, Natalia I Denisenko, Michale S Fee\***

McGovern Institute for Brain Research, Department of Brain and Cognitive Sciences, MIT, Cambridge, United States

**Abstract** Behaviors emerge via a combination of experience and innate predispositions. As the brain matures, it undergoes major changes in cellular, network, and functional properties that can be due to sensory experience as well as developmental processes. In normal birdsong learning, neural sequences emerge to control song syllables learned from a tutor. Here, we disambiguate the role of tutor experience and development in neural sequence formation by delaying exposure to a tutor. Using functional calcium imaging, we observe neural sequences in the absence of tutoring, demonstrating that tutor experience is not necessary for the formation of sequences. However, after exposure to a tutor, pre-existing sequences can become tightly associated with new song syllables. Since we delayed tutoring, only half our birds learned new syllables following tutor exposure. The birds that failed to learn were the birds in which pre-tutoring neural sequences were most 'crystallized,' that is, already tightly associated with their (untutored) song.

## Editor's evaluation

This fundamental work in songbirds shows that stereotyped neural sequences known to drive the correspondingly stereotyped acoustic structures of adult songs can exist very early in development even when songs are variable and before birds have been provided song models by tutors. The evidence is exceptional and includes imaging activity of populations of premotor neurons in singing birds. This paper provides important insights into the mechanistic foundations of how nature and nurture work together to produce learned motor sequences.

**\*For correspondence:**
fee@mit.edu

**Competing interest:** The authors declare that no competing interests exist.

## Introduction

On the one hand, sensory experience is known to be essential for the normal development of brain circuits. On the other hand, genetically specified developmental processes are also essential – we learn too quickly and from too sparse data to rely on sensory experience alone (*Chomsky et al., 2002*). Thus, it appears that the brain is able to use genetically specified predispositions to fill in gaps in its sensory experience. When typical sensory experience is absent or delayed, certain aspects of brain development proceed anyway, while other aspects are delayed. This is true both in primary sensory systems (*Wiesel and Hubel, 1963*; *Bear and Singer, 1986*; *Farley et al., 2007*; *Ye et al., 2021*), and for more cognitive behaviors such as social interaction and language (*Hildyard and Wolfe, 2002*; *Moreno-Torres et al., 2016*; *Kral et al., 2019*). Brain circuits acquire structure and organization even in the absence of typical training inputs. Here, we examine this self-organized structure, and what happens when the sensory experience of a conspecific tutor is reintroduced, in the context of songbird vocal learning.

Song learning is influenced by both auditory exposure to a particular tutor song, and by inherited preferences (*Tchernichovski and Marcus, 2014*). It is well known that songbirds, in the absence of exposure to a tutor bird, develop 'isolate' songs, with highly variable and atypical syllable rhythms

(*Price, 1979*; *Williams et al., 1993*; *Fehér et al., 2009*). However, when these 'isolate' songs are used as tutor songs, after two generations birds sing normally again, suggesting that an 'innate' preference filters what aspects of a tutor song are actually imitated (*Fehér et al., 2009*). Song imitation requires remarkably little total exposure to a tutor song – approximately 75 s total on a single day is enough for a bird to remember a song, and subsequently practice and imitate it (*Deshpande et al., 2014*). Zebra finches, like many songbird species, are able to imitate songs of birds from other species, but when given a choice they prefer zebra finch songs (*Eales, 1987*). Furthermore, inherited genetic predispositions have a strong effect on both the precise tempo at which a zebra finch sings its song (*Mets and Brainard, 2018*), as well as the particular learning styles of individual birds (*Mets and Brainard, 2019*). Thus, within the songbird brain, we expect to see an interplay between developmentally specified and learned structure.

There are several possibilities for what happens in the brain during isolate song, and how it compares to typical (tutored) brain development. In typical birds, neurons in HVC are initially only weakly coupled to a song, firing only at the onsets of syllables when birds are babbling subsong (*Okubo et al., 2015*), and HVC is not necessary for subsong production (*Aronov et al., 2008*). Then, as the song becomes more mature and repeatable, each HVC projection neuron fires at its own precise moment during the song, with neurons firing one after the other, together forming a stable sequence of neural firing that tiles the song (*Okubo et al., 2015*; *Lynch et al., 2016*; *Picardo et al., 2016*), in interplay with inhibitory neurons (*Kosche et al., 2015*; *Vallentin et al., 2016*). This maturation process in HVC has been modeled as an initially random network of neurons that, with the right training inputs and plasticity rules, assembles into a chain of sequentially connected neurons (*Buonomano, 2005*; *Jun and Jin, 2007*; *Fiete et al., 2010*; *Okubo et al., 2015*; *Figure 1A*). However, what happens in birds isolated from a tutor? Compared to typical adult zebra finch songs, isolate song has a much less stable sequence of syllables and abnormally variable acoustic structure and timing (*Fehér et al., 2009*). In fact, aspects of isolate songs resemble features of early babbling (subsong). Does HVC in isolate birds resemble that of subsong birds? Or does HVC mature to form sequences, even without the experience of a tutor, and without the behavioral stereotypy seen in adult birds? We use functional calcium imaging in singing isolated birds to address these questions.

By observing the neural activity in the HVC of isolated birds, we found that the HVC network activity can mature into long repeatable sequences even without exposure to a tutor. However, there are some key differences between typical adult HVC sequences and those found in isolated birds, suggesting which features of HVC development rely on exposure to a tutor. Next, we observe HVC in isolated birds immediately before and after delayed exposure to a tutor. Birds isolated from a tutor are able to learn a song if exposed to a tutor before the end of a critical period, typically around age 65 days post-hatch (dph), but are increasingly unable to learn at later ages (*London, 2019*; *Gobes et al., 2019*; *Immelmann, 1969*). Although only half of our late tutored birds successfully learned from the tutor, we observed an interesting correlation between HVC activity prior to tutoring and the degree to which birds learned. Namely, birds with highly song-locked HVC activity prior to tutoring typically failed to learn, while birds with less song-locked activity tended to learn. In the birds that did learn, we were able to track sequences throughout the course of learning. Pre-existing self-organized HVC sequences persisted throughout major changes to the song, forming a substrate for newly learned song elements. Together, these results point to how the brain may self-organize, and at the interplay between self-organized structure and the ability to incorporate new information from a tutor.

## Results
### Neural sequences are present in isolated birds, but atypical
We first asked whether the songs of isolated birds involve the same pre-motor neural pathways and neuronal sequences responsible for generating typical songs. We carried out functional calcium imaging of large populations of neurons in HVC of isolated birds at a range of ages. Sequences of neuronal activity in HVC have previously been analyzed by aligning neuronal activity to repeatable elements of the song (*Long et al., 2010*), an approach with limited utility in isolated birds due to the high variability of their songs. Instead, we extract neural sequences directly from the calcium signals using an unsupervised algorithm, seqNMF, (*Mackevicius et al., 2019*) to find the sequences that best fit the neural data.

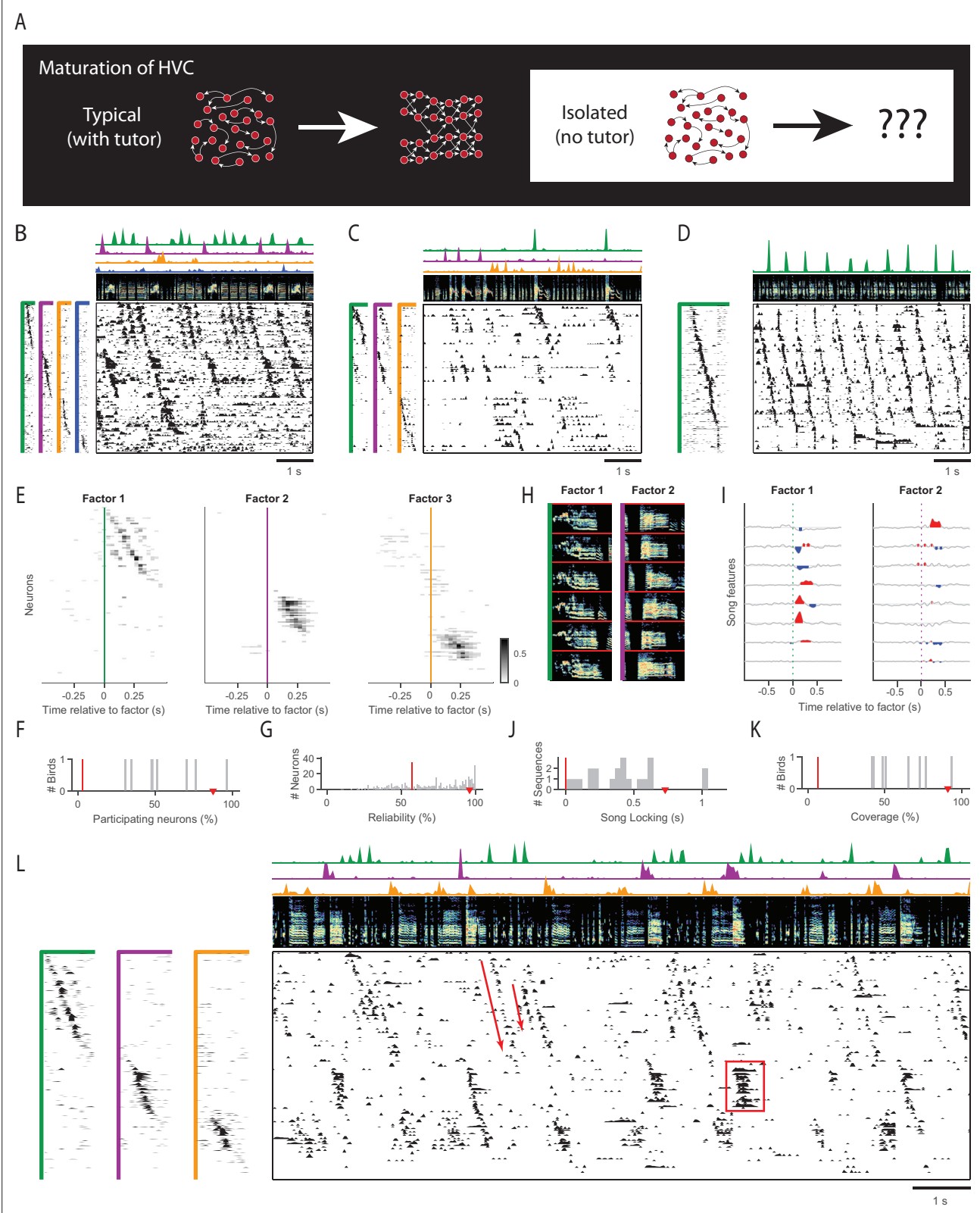

**Figure 1.** Sequences in isolated birds. (**A**) Diagram of HVC maturation. In typically tutored birds, HVC sequences appear to grow and differentiate over time. (**B**) Example neural sequences recorded in a singing isolated bird (older juvenile, 61 dph). Main panel (lower right), functional calcium imaging recordings from 98 neurons. Rows (neurons) are sorted according to sequences (factors) extracted by unsupervised algorithm seqNMF (see Methods). (Above) Song spectrogram (0–10kHz). The four sequence factor exemplars and timecourses are shown to the left and above, in corresponding colors.

*Figure 1 continued on next page*

*Figure 1 continued*

Duration of factor exemplars: 0.5s. (**C**) Same as B, for another example isolated bird (adult, 117 dph). (**D**) Same as in B, for a typically tutored bird (adult, 217 dph). (**E**) Time-lagged cross-correlation between each neuron and each of the three extracted factors recorded in a singing isolated bird (older juvenile, 68 dph). Only significant bins in the cross correlation are shown (p<0.05, Bonferroni corrected, compared to a circularly-shifted control). (**F, G, J, K**) Sequence properties in isolated birds. For reference, the median for a typically tutored bird in D is shown by a red triangle, and the median for a control dataset where each row was circularly shifted by a random amount is shown by a red line. (**F**) Percent of neurons participating in at least one extracted sequence. (**G**) Reliability of participating neurons across sequence renditions. Note that in the control dataset, relatively few neurons participate. (**H**) Example song spectrograms (0.5s) extracted at moments when neural sequences were detected in an isolated bird (older juvenile, 64 dph) (**I**) Correlation of these sequences with eight song features (top to bottom: amplitude, entropy, pitch goodness, aperiodicity, mean frequency, pitch, frequency modulation, amplitude modulation). (**J**) Strength of song locking (see Methods). (**K**) Percent of the song covered by some sequence. (**L**) Example of sequence abnormalities in an isolated bird (same as in E). Sequences of inconsistent length (8/8 isolated birds) and ensemble persistent activity (7/8 isolated birds) are annotated in red.

The online version of this article includes the following figure supplement(s) for figure 1:

**Figure supplement 1.** HVC sequences exist even in young isolated birds.

**Figure supplement 2.** Estimating the number of significant sequences in each dataset.

**Figure supplement 3.** Enlarged data, bird 6961.

**Figure supplement 4.** Enlarged data, bird 6991.

**Figure supplement 5.** Enlarged data, bird 7030.

**Figure supplement 6.** Enlarged data, bird 7187.

**Figure supplement 7.** Enlarged data, bird 6938.

**Figure supplement 8.** Enlarged data, bird 6992.

**Figure supplement 9.** Enlarged data, bird 6962.

**Figure supplement 10.** Enlarged data, bird 6922.

The seqNMF algorithm is designed to find patterns in data, and identify the times when each pattern occurs. It is a generalization of non-negative matrix factorization, allowing for non-synchronous patterns of neural activity, for example, sequential firing of neurons in a chain. In the context of seqNMF, a sequence is defined as a pattern of activity in a population of neurons that extends for more than one time step. The algorithm allows for an individual neuron to be active at more than one time in a sequence, but in practice neurons in HVC or the hippocampus are typically active only once in a sequence. Each seqNMF factor is represented as an exemplar sequence $W$, and a vector of times when the sequence occurs, $H$. A sequence in the data is factorized as the convolution of a $W$ with its corresponding $H$. Data often contain multiple sequences, each represented by different $W$s and $H$s, and the full data matrix is represented as a sum across all different extracted sequences. Using gradient descent, the algorithm automatically finds sequences that best fit the data, while avoiding redundancies between sequences. It allows for explicitly selecting between 'parts-based' and 'events-based' factorizations, but here we are agnostic to this distinction, selecting the factorization that best fits the data. SeqNMF has been shown to be robust to several forms of neural variability (*Mackevicius et al., 2019*). In addition, it includes a measure of 'sequenciness,' which quantifies the extent to which data are better explained by temporally extended sequences than by synchronized activity. This measure ranges from zero (only synchronized activity) to one (only temporally extended sequences, such that shuffling timebins in the data erases all repeatable structure).

This technique reveals the existence of significant sequential activity in HVC of singing isolated birds (*Figure 1B and C*, data in C also presented in *Mackevicius et al., 2019*). In these birds, the seqNMF factorization explained 27 ± 9% of the power in the neural data, with a sequenciness score of 0.55 ± 0.88 (mean ± standard deviation). SeqNMF also reveals long continuous sequences in data acquired from typical adult HVC (*Figure 1D*) as expected from previous work (*Long et al., 2010*; *Picardo et al., 2016*; *Lynch et al., 2016*). Here, 46% of the data is explained by the seqNMF factorization, with sequenciness score of 0.74.

The sequences found in isolated birds are surprisingly typical in some respects, but atypical in others, especially in their correlation to vocal output. As in typical HVC sequences, neurons in isolated birds participate at characteristic moments during the sequence (*Figure 1E*), and many neurons participate in at least one sequence (*Figure 1F*). Neurons that participate in a sequence tend to fire at a majority of sequence occurrences (*Figure 1G*). On average, 45.21 ± 7% of these neurons' activity

was specific to sequences. Neural sequences are correlated with precisely timed song features in isolated birds' songs (*Figure 1H1*, song features calculated as in *Tchernichovski et al., 2000*). The extent of song locking is quantified by computing the cross-correlation between sequences and song features, and summing across significant time lags (see Methods). Song locking in isolated birds was only on average 0.58 times as strong as in a typically tutored adult bird (*Figure 1J*). Finally, in isolated birds, on average only 61% of each song bout is represented by a detected HVC sequence, substantially less than the complete sequence coverage found in typically tutored birds (*Picardo et al., 2016*; *Lynch et al., 2016*; *Okubo et al., 2015*; *Figure 1K*, see Methods). By each of these measures, HVC sequences in isolated birds are more strongly present than would be expected from per-neuron circularly shifted data (red lines in *Figure 1F–K*), but not as robust as sequences observed in typically tutored birds (red triangles in *Figure 1F–K*). The per-neuron circular shifting controls for the possibility that neurons in isolate HVC may be firing independently. For enlarged panels of pre-tutoring neural and song data from each of the eight isolated birds, see *Figure 1—figure supplements 3–10*.

HVC activity in isolated birds exhibits additional qualitative differences from that in typically tutored birds. While HVC neurons generate only brief bursts of spikes in tutored birds, neurons in isolated birds sometimes generated extended periods of continuous activity, especially during long syllables of variable duration (*Figure 1L*, 7/8 birds exhibited multiple instances of persistent activity, coordinated across at least three neurons, and lasting at least 500 ms). This contrasts with the long syllables of typical adult songs which are all generated by extended sequences of brief bursts. In addition, HVC sequences in isolated birds exhibit variable durations, often truncating at different points (*Figure 1L*, 8/8 birds), producing syllables of highly variable duration. Such truncations in the middle of a syllable, as opposed to at stereotyped gaps between syllables, are very unusual in typically tutored birds (*Cynx, 1990*). These atypical modes of HVC activity suggest several possible mechanisms to understand characteristic features of isolate songs, abnormally long syllables, and those of variable duration (*Fehér et al., 2009*). For example, syllables in isolated birds may exhibit variable duration when their underlying HVC sequences are truncated at different points. We wondered if the existence of sequences in HVC of socially isolated birds occurs only after the closure of the critical period (i.e. a product of an already atypical isolate song) or whether they develop at an even earlier age when birds have not yet heard a tutor song, but can still be tutored. We recorded in 5 birds at ages 57–64 dph, prior to tutor exposure, and found strong evidence for HVC sequences (*Figure 1—figure supplement 1A*). In each bird, HVC data exhibited significantly higher sequenciness scores than time-shuffled control datasets ($p < 0.05$). There was not a significant correlation between the age of the bird and any sequence features we measured (*Figure 1—figure supplement 1B-F*, linear regression model, significance threshold p<0.5, compared to a constant model). The correlations were not significant both when we restricted to birds within the traditional critical period ($lt_{65}$ dph), and when we included data from three older isolated birds (68–117 dph). Thus, the large (several fold) bird-to-bird variability in sequence properties (*Figure 1—figure supplement 1B-F*) is not explained by age, and is likely due to inter-individual variability in developmental timecourses.

## Prior to tutoring, birds that will learn exhibit HVC sequences that are relatively immature and decoupled from vocal output

Next, we asked whether the properties of the HVC sequences relate to the ability of birds to learn a new song from a tutor. Many of our young isolated birds were eventually tutored at an age around the critical period and we found that half of them learned elements of their tutor song, while the others developed fully isolated songs. We classified birds as learners if their song had an Imitation Score metric (*Mandelblat-Cerf and Fee, 2014*) greater than 0.5. The songs of non-learners remained highly variable and isolate-like even after tutoring (*Figure 2A*). In contrast, learner birds developed a new syllable within a day or two after tutoring, and ultimately sang typical adult songs, consisting of stereotyped motifs (*Figure 2B*). The new syllables appeared to imitate elements of the tutor song, and persist as part of the learned song (*Figure 2—figure supplements 1–4*).

An analysis of HVC activity revealed that sequences prior to tutoring were systematically less mature/'crystallized' in birds that learned than in birds that failed to learn. Learner birds had fewer sequences than non-learners (*Figure 2C*, average two sequences in learners, 3.25 sequences in non-learners, p=0.029, Wilcoxon rank sum test). Sequences in learner birds were more weakly correlated to song features (*Figure 2D*, average 0.20 s learner, 0.55 s non-learner, p=0.0034, Wilcoxon rank sum

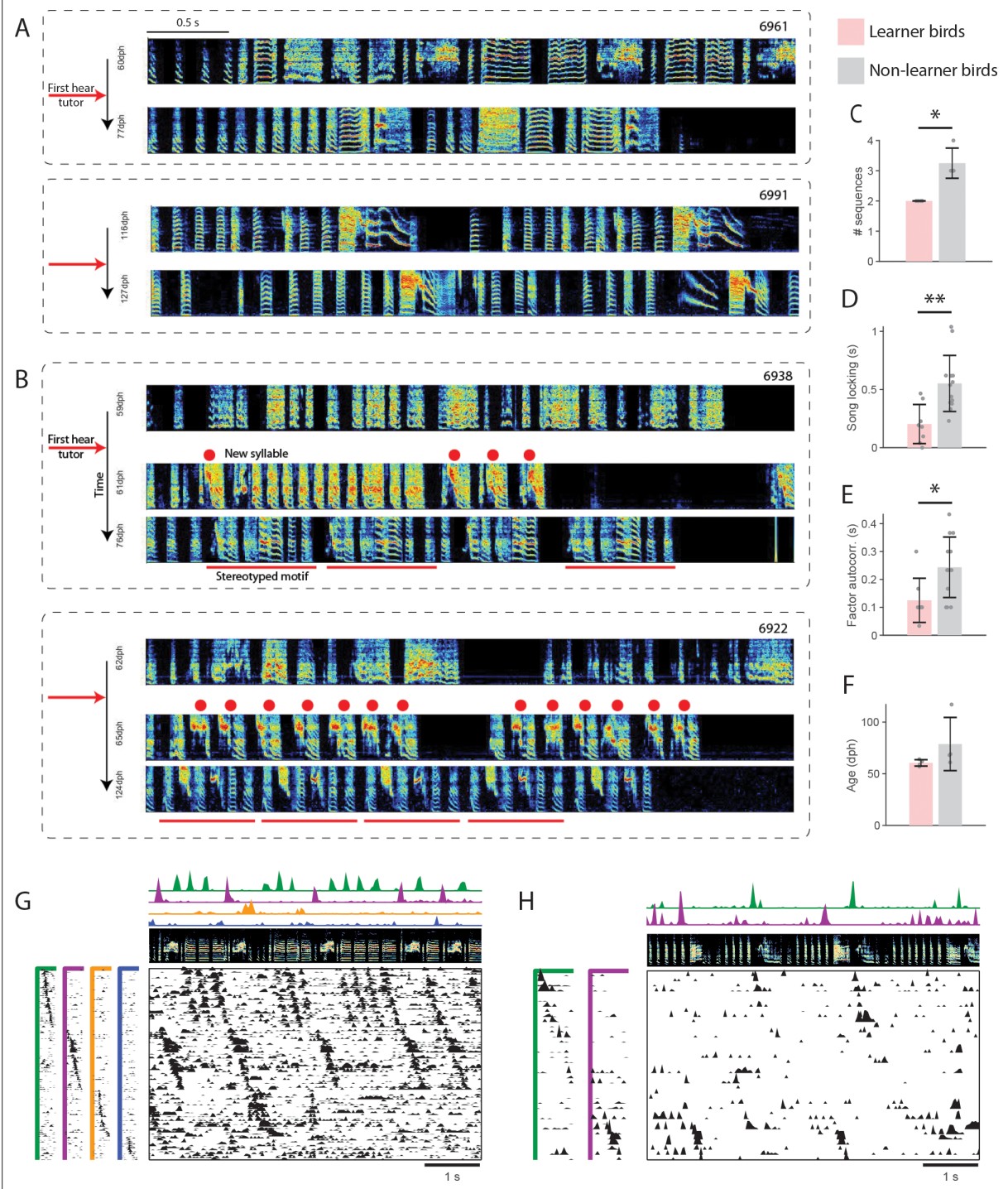

**Figure 2.** Relation between HVC sequence maturity and subsequent song learning. (**A**) Example spectrograms for two non-learner birds, prior to tutoring and several weeks later (at least 77dph). (**B**) Example spectrograms for two learners, prior to tutoring, shortly after tutoring, and several weeks later. Red dots mark the new syllable. Red bars mark a stereotyped motif. (**C–E**) Three measures of HVC sequence maturity for learners (pink) and non-learners (gray). Error bars denote standard deviation (*p<0.05, *p<0.01). (**C**) Number of sequences in HVC. (**D**) Fraction of neurons that participate in a sequence. (**E**) Autocorrelation of sequence factor timecourses. (**F**) Age of first tutoring for learners and non-learners. (**G–H**) Example pre-tutoring data from two birds that were brothers. (**G**) A non-learner first tutored at 61 dph. (**H**) A learner first tutored at 64 dph.

The online version of this article includes the following figure supplement(s) for figure 2:

**Figure supplement 1.** Elaborations on tutor match, bird 6938.

**Figure supplement 2.** Elaborations on tutor match, bird 6992.

*Figure 2 continued on next page*

*Figure 2 continued*

**Figure supplement 3.** Elaborations on tutor match, bird 6962.

**Figure supplement 4.** Elaborations on tutor match, bird 6922.

test). Sequences in learners had lower autocorrelation, a measure of how repeatably/rhythmically they are produced (*Okubo et al., 2015*), than non-learners (*Figure 2E*, average 0.125 s learner, 0.244 s non-learner, p=0.018, Wilcoxon rank sum test). Three additional measures of sequence maturity, all related to intrinsic sequence properties were calculated. While non-learners also trended higher in these measures, the differences were not significant (Wilcoxon rank sum tests, Neural participation: average 45% learner, 70% non-learner, p=0.2; Reliability: average 69% learner, 74% non-learner, p=1; Coverage: average 51% learner, 71% non-learner, p=0.34). In our dataset, the age of tutoring did not significantly influence whether the bird was a learner or non-learner (*Figure 2F*, average 60.5 dph learner, 78.75 dph non-learner, p=0.11, Wilcoxon rank sum test). For example, one of the younger birds in our dataset (61 dph) was a non-learner, and had particularly clear HVC sequences before tutoring (*Figure 2G*). This bird's brother, tutored 3 days later, was a learner, and had sequences that appear far less mature (*Figure 2H*). Together, these results suggest that the presence, at the time of tutoring, of robust song-locked sequences, may inhibit learning. In other words, learning may be better supported by more immature sequences that are more independent from the vocal output.

## Tracking HVC sequences across rapidly learned song changes

In late-tutored birds that learned, the speed with which new syllables appeared was striking. These birds developed a new syllable within a day or two after tutoring (*Figure 2A and B*), as has been previously described (*Tchernichovski et al., 2001*; *Lipkind and Tchernichovski, 2011*; *Lipkind et al., 2013*). These new syllables appeared to emerge de-novo, not by syllable differentiation as is common in tutored birds.

We wondered if these birds, which learned a new syllable rapidly after tutoring, formed a de-novo HVC sequence for this new syllable, or perhaps used a pre-existing sequence. We were able to track neurons in our calcium imaging data throughout the course of tutoring (*Figure 3A*, see Methods, *Gu et al., 2023*), enabling us to see what happens to neural activity during rapid changes in the song. We first extracted neural sequences associated with new post-tutoring syllables, then followed these neurons back in time to find that the sequence existed even prior to tutoring (*Figure 3B and C*, see Methods). The correlation coefficients between the pre-tutoring and post-tutoring sequences in the four learner birds were 0.20, 0.38, 0.27, and 0.45, respectively. In all four learner birds, the correlation between the pre-tutoring and post-tutoring sequences was significantly higher than expected by a time-shuffled control (p=0.014, p=0.018, p=0.0004, p=0.0002, respectively). Additionally, the correlation between pre- and post- tutoring sequences was significantly higher than a neuron-shuffled control in two birds (p=0.0099, p=0.0032), and trended higher in the remaining two birds (p=0.077, p=0.079, likely influenced by an abundance of near-synchronous neurons, *Figure 3C*). Interestingly, the sequences prior to tutoring were relatively uncoupled to vocal output, without a strong correlation to song syllables. Combining data from the four birds that learned a new syllable rapidly after tutoring, neural sequences extracted two days after tutoring appeared to become more song locked after tutoring (*Figure 3D*, p=0.0048, Wilcoxon rank sum test). This lack of coupling between HVC sequences and song structure prior to tutoring, and their subsequent incorporation into learned syllables, suggests that, in these late-tutored birds, HVC sequences may exist in a 'latent' state.

Next, we aimed to control for the possibility that the appearance of sequences becoming progressively more locked to vocal output after tutoring was due to the fact that sequences were extracted from neural data recorded after tutoring. We directly extracted HVC sequences from exclusively pre-tutoring neuronal data and tracked them forward in time until a new syllable appeared. Sequences that were initially relatively 'latent' persisted, becoming progressively more correlated with vocal output, ultimately appearing to become locked to a new syllable (*Figure 3E*). Each of the 'learner' birds appeared to have two HVC sequences present prior to tutoring. Of these sequences, the ones that started off less correlated with song amplitude exhibited a significant increase in correlation with song amplitude after tutoring (*Figure 3F*, p=0.045, Wilcoxon rank sum test). The sequences that started off more correlated with song amplitude did not significantly change their correlation with song amplitude (*Figure 3G*, p=1, Wilcoxon rank sum test). Together, these results are consistent

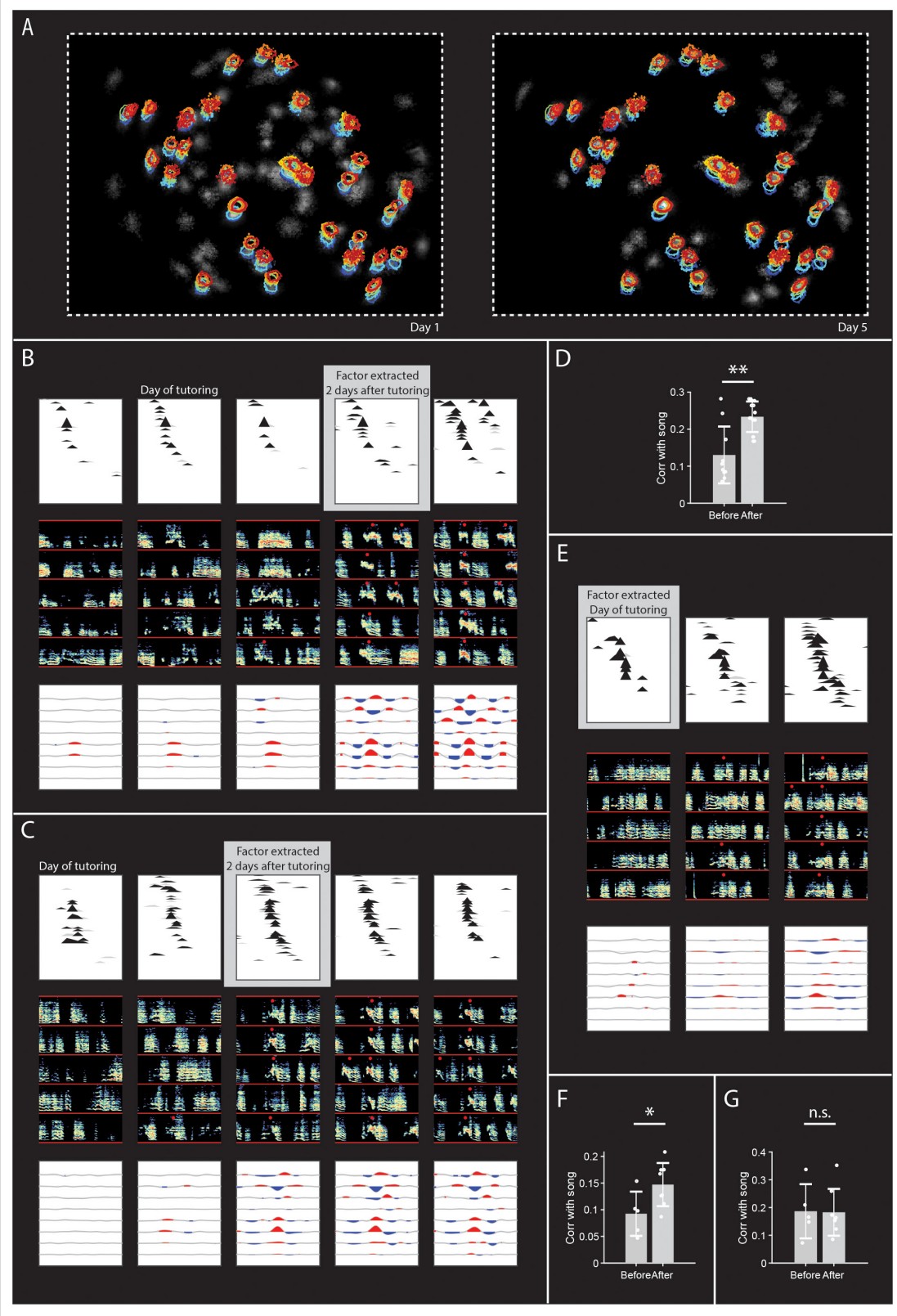

**Figure 3.** Tracking HVC sequences as isolated birds rapidly learn a new syllable. (**A**) Neurons detected before (left) and after (right) tutoring shown in grayscale (CNMF_E algorithm). Colored contours indicate locations of neurons tracked across five days, from blue to red (**B**) Sequence in HVC, tracked before and after first tutor exposure, through the development of a new syllable. Each of the five panels shows data from a different day, starting one day before tutoring, through three days after tutoring. (Top) On each recording day, cross-correlation of neurons with the sequence that becomes

*Figure 3 continued on next page*

*Figure 3 continued*

associated with the new syllable. The sequence was extracted by running seqNMF on neural data recorded two days after tutoring, and selecting the factor associated with the new syllable. Neurons are sorted according to participation in this factor. Significant bins are shown in black, non-significant bins in gray (p=0.05, Bonferroni corrected, compared to circularly-shifted control) (Middle). On each recording day, for example, spectrograms at times when the sequence occurs on each day. Red circles indicate putative newly learned syllables. (Bottom) On each recording day, cross-correlation of sequence with acoustic features (amplitude, entropy, pitch goodness, aperiodicity, mean frequency, pitch, frequency modulation, and amplitude modulation) (**C**) Same as B for a different example bird. (**D**) Correlation with song amplitude before (pink) and after (gray) tutoring for all sequences in learner birds extracted data when a new syllable had been learned. (**E**) Similar to B and C, for a different example bird. Here, sequences are extracted from pre-tutoring data, then tracked forward in time. (**F**) Song locking (maximum cross-correlation with song amplitude) before and after tutoring for the pre-tutoring sequences that had weaker song locking. (**G**) Song locking before and after tutoring for the pre-tutoring sequences that started off with stronger song locking.

with the view that the emergence of new syllables after tutoring may co-opt existing HVC sequences, including relatively 'latent' sequences.

## Discussion

We set out to determine whether the formation of sequences in HVC depends on prior exposure to a tutor song. By observing the neural activity in HVC of isolated birds, we found that HVC network activity can form long repeatable sequences even in birds that had no prior exposure to vocal tutoring. Sequences in isolate HVC exhibit some properties of typical HVC, with many neurons reliably participating in sequences, and sequences being correlated to vocal output. However, sequences in isolated birds were less reliable and less tightly correlated with vocal output than has been described in typical birds, and exhibited abnormal truncations and persistent activity.

We had previously hypothesized that the experience of hearing a tutor may seed the formation of HVC sequences of the appropriate number and durations (*Mackevicius and Fee, 2018*), but our new data reveal that HVC sequences exist even prior to tutoring. Thus, there must be a way for sequences to form without the prior storage of a tutor memory. In models of Hebbian learning in HVC, sequences can form in networks driven by random inputs rather than patterned inputs (*Fiete et al., 2010*). However, in this case, the distribution of sequence durations no longer matches syllable durations found in typical adult birds, but is instead more consistent with the highly variable and atypically long syllables that occur in birds that have never heard a tutor (isolate song) (*Fehér et al., 2009*; *Price, 1979*). Thus our findings are consistent with the view that sequences can emerge in isolate birds by a combination of simple Hebbian learning mechanisms together with spontaneous activity either within HVC or driven by the inputs to HVC.

Our discovery of latent sequences suggests a separation between neural processes for building a stable representation of states within a task (i.e. sequential moments in time), and neural processes for associating an action with each state. Thus, sequences may gradually emerge in the maturing HVC network via simple Hebbian processes (*Buonomano, 2005*; *Jun and Jin, 2007*; *Fiete et al., 2010*; *Okubo et al., 2015*), but may remain relatively decoupled from downstream motor neurons until a memory of the tutor song is learned and reinforcement learning processes begin.

From a computational perspective, what do latent sequences tell us about how the brain learns? By latent sequences, we mean sequences that are initially only weakly correlated with vocal output, but are subsequently used to produce learned song changes. In reinforcement learning models of song learning, HVC sequences remain relatively stable even as the song changes (*Doya and Sejnowski, 1994*; *Fiete et al., 2007*; *Fee and Goldberg, 2011*; *Brainard and Doupe, 2013*), consistent with our observation of stable sequences. This is in contrast with other models of song learning, like the 'inverse model' (*Giret et al., 2014*; *Hanuschkin et al., 2013*; *Hahnloser and Ganguli, 2013*). In the inverse model, each motor neuron produces the same vocal output at different times during vocal learning; song changes are caused by pre-motor neurons (e.g. HVC) being activated in a different order. In contrast, we observed relatively stable sequences throughout learned song changes. Our results are consistent with data from the primary motor cortex of macaques operating a brain-computer interface—a fixed repertoire of activity patterns is associated with different movements after learning (*Golub et al., 2018*). Our results are also consistent with the idea that the brain may use pre-existing sequential patterns to rapidly learn from new experiences, for example, the existence of sequences in

the hippocampus prior to exposure to new environments (*Villette et al., 2015*; *Farooq et al., 2019*; *McKenzie et al., 2021*; *Dragoi, 2020*).

If the brain is able to build on the latent structure to learn from sparse data, essentially implementing inductive bias, we might expect different forms of latent structure for different tasks. Zebra finches are known to develop typical songs, including typical syllable durations, after being tutored by atypical isolate songs, relying on species-specific 'priors' to achieve species-typical syllable durations. The latent sequences we observed tended to last approximately a couple hundred milliseconds—the same as the duration of typical zebra finch syllables. Might other species that sing faster songs (e.g. grasshopper sparrow) or slower songs (e.g. white-throated sparrow) exhibit latent sequences of shorter or longer durations? One might imagine that the speed of latent sequences could be genetically specified by expression levels of ion channels with different time constants within HVC. Alternatively, the duration of latent sequences could be specified by the amount of time it takes for HVC to get feedback from respiratory and/or auditory centers, which may also have their own intrinsic timing or rhythmicity (*Schmidt and Goller, 2016*; *Hamaguchi et al., 2016*; *Araki et al., 2016*; *Mackevicius et al., 2020*). Each of these possible sources of latent HVC structure could be tested in further experiments. By whatever mechanism latent sequences arise, they appear to be capable of supporting song learning, at least in the case of delayed tutoring. More generally, the ability of brains to generate complex learned behavior may depend on the intrinsic developmental formation of appropriate latent dynamics in motor and sensory circuits.

# Materials and methods

**Key resources table**

| Reagent type (species) or resource | Designation | Source or reference | Identifiers | Additional information |
|---|---|---|---|---|
| Software, algorithm | seqNMF | *Mackevicius et al., 2019* | github.com/FeeLab/seqNMF | Sequence detection |
| Software, algorithm | CNMF_E | *Zhou et al., 2018* | github.com/zhoupc/CNMF_E | Cell extraction |
| Software, algorithm | STAT | *Gu et al., 2023* | https://github.com/shijiegu/STAT | Tracking neurons across days |
| Software, algorithm | Chronux | *Mitra and Bokil, 2007* | chronux.org/ | Spectrogram computation |
| Software, algorithm | SAP | *Tchernichovski et al., 2000* | soundanalysispro.com | Sound analysis |
| Software, algorithm | SI | *Mandelblat-Cerf and Fee, 2014* | https://doi.org/10.1371/journal.pone.0096484 | Song imitation |
| Software, algorithm | MATLAB | MathWorks | mathworks.com/products/matlab | Programming language |
| Biological sample (*Taeniopygia guttata*) | Zebra finches | MIT animal facility | *Taeniopygia guttata* | |
| Strain, strain background (*adeno-associated virus*) | AAV9.CAG.GCaMP6f.WPRE.SV40 | *Chen et al., 2013* | Addgene viral prep # 100836-AAV9, http://n2t.net/addgene:100836, RRID:Addgene_100836 | |
| Commercial assay or kit | Miniature microscope | Inscopix | https://www.inscopix.com/nvista | |

## Animal care and use

For this study, Imaging data were collected in nine male zebra finches (*Taeniopygia guttata*) from the MIT zebra finch breeding facility (Cambridge, MA). Animal care and experiments were carried out in accordance with NIH guidelines, and reviewed and approved by the Massachusetts Institute of

Technology Committee on Animal Care (Protocol 0721-064-24: Chronic Recording of Neural Activity in Songbirds).

In order to control exposure to a tutor song, eight birds were foster-raised by female birds, which do not sing, starting on or before post-hatch day 15 (15 dph). Starting between 35 dph and 50 dph, these birds were housed singly in custom-made sound isolation chambers. An additional bird was tutored by his father, as is typical. After a couple of days of acclimation to the lab environment, birds were anesthetized with isoflurane, and were given a surgery to inject the virus to express the functional indicator GCaMP6f and implant a GRIN (gradient index) lens (see below). Analgesic (Buprinex) was administered 30 min prior to the surgery, and for 3 days postoperatively. After at least a week for virus expression, an Inscopix miniscope baseplate was attached to the existing implant. Birds were acclimated to the miniscope, and once birds started singing with the miniscope (2–7 days), functional calcium signals were recorded for at least three consecutive days. To avoid photobleaching, short files (approximately 10 s) were obtained, typically fewer than 50 files per day. Once some pre-tutoring singing data had been obtained, birds were tutored briefly (5–10 song bouts from a tutor bird) each day for at least 7 days.

## Expression of functional calcium indicator GCaMP6f

The calcium indicator GCaMP6f was expressed in HVC by intercranial injection of the viral vector AAV9.CAG.GCaMP6f.WPRE.SV40 (*Chen et al., 2013*) into HVC. In the same surgery, a cranial window was made using a relay GRIN (gradient index) lens (1 mm diameter, 4 mm length, Inscopix) implanted on the surface of the brain, after the dura was removed. After at least one week, in order to allow for sufficient viral expression, recordings were made using the Inscopix nVista miniature fluorescent microscope. It is not known whether this virus is specific to projection neurons in HVC. However, given the dense firing patterns of HVC interneurons, once convolved with the GCaMP6f calcium kernel, HVC interneuron activity would likely look flat, almost indistinguishable from the background. Therefore, in practice, most if not all of the neurons included in our datasets are likely to be projection neurons, not HVC interneurons.

## Extraction of neuronal activity and background subtraction using CNMF_E

Neuronal activity traces were extracted from raw fluorescence movies using a constrained non-negative matrix factorization algorithm, CNMF_E, that is specialized for microendoscope data by including a local background model to remove activity from out-of-focus cells (*Zhou et al., 2018*). Custom software (Shijie Gu, Emily Mackevicius, Pengcheng Zhou) was used to extend the CNMF_E algorithm to combine batches of short files (BatchVer) and track individual neurons over the course of multiple days (*Gu et al., 2023*).

## Unsupervised discovery of neural sequences using seqNMF

We addressed the challenge of needing to detect neural sequences in HVC without relying on aligning neural activity to the song by developing an unsupervised algorithm, seqNMF (*Mackevicius et al., 2019*). This was necessary because juvenile songs are highly variable and difficult to parse into repeatable syllables, and because we wanted to allow for the possibility that HVC activity might be more stereotyped than the song. Briefly, seqNMF factorizes data into exemplar sequence factors ($W$'s). Each sequence factor has a corresponding timecourse ($H$). Convolving each exemplar with its respective timecourse produces an approximate reconstruction of the original data ($\widetilde{X} = W \circledast H$). SeqNMF returns a factorization that minimizes reconstruction error, subject to a penalty term that encourages simpler factorizations. It is possible for seqNMF to extract non-sequential (purely synchronous) patterns of neural activity. SeqNMF measures the extent to which a dataset contains sequential vs. synchronous patterns using a 'sequenciness' score, which ranges between 0 and 1. A score of zero indicates that all explanatory power of the factorization is due to synchronous activity, while a score of one indicates that none of the explanatory power is due to synchronous activity. Quantitatively this is calculated by shuffling the timebins in the dataset. In these datasets, both 'sequenciness' scores and visual inspection of the factors suggest the presence of temporally extended patterns, which we refer to as sequences.

### Preprocessing calcium traces prior to running seqNMF

We performed several preprocessing steps before applying seqNMF to functional calcium traces extracted by CNMF_E. First, we estimated burst times from the raw traces by deconvolving the traces using an AR-2 process. The deconvolution parameters (time constants and noise floor) were estimated for each neuron using the CNMF_E code package (*Zhou et al., 2018*). Some neurons exhibited larger peaks than others, likely due to different expression levels of the calcium indicator. Since seqNMF would prioritize the neurons with the most power, we renormalized by dividing the signal from each neuron by the sum of the maximum value of that row and the $95^{th}$ percentile of the signal across all neurons. In this way, neurons with larger peaks were given some priority, but not much more than that of neurons with weaker signals.

### Estimating the number of significant sequences in each dataset

The number of sequences present in real neuronal datasets can be slightly ambiguous, so we used several methods to arrive at and validate an estimate for the number of significant neural sequences present in each dataset. It is important to note that, since our datasets are short, there may be additional neural sequences in HVC that do not appear, or do not achieve significance, in our datasets.

Our primary method for estimating the number of sequences in each dataset involves choosing a value for the seqNMF parameter $\lambda$ that balances reconstruction cost with correlation (redundancy) cost, as described in *Mackevicius et al., 2019*. We swept $\lambda$ with $K = 10$ and $L = 0.5$ s to find $\lambda_0$, the cross-over point that balances these cost terms (*Figure 1—figure supplement 2A*). Based on analysis of simulated data (*Mackevicius et al., 2019*), where values of $\lambda$ at or slightly above $\lambda_0$ yielded the correct number of significant sequences, we looked at the distribution of significant sequences at $\lambda = \lambda_0$ and $\lambda = 2\lambda_0$ (*Figure 1—figure supplement 2B*), and chose as our estimate a number between the peaks of these two distributions. For significance tests, each dataset was split into a training set (75%) and a test set (25%). Sequences were detected in the training dataset, and significance was measured in the test dataset by assessing the overlap of the sequences with the test dataset compared to null (time-shifted) sequences.

We validated these estimates of the number of significant sequences by testing the consistency of resulting factorizations. Consistency measures the extent to which there is a one-to-one mapping between the factors of two different factorizations. When K matches the number of sequences in a dataset, seqNMF factorizations will be relatively consistent across different random initializations, even at $\lambda = 0$ (*Mackevicius et al., 2019*). We ran seqNMF on the entire dataset at the estimated K from 25 different random initial conditions, and confirmed that the sequences were consistent across the different initializations. This can be seen by a large block of consistent factorizations in the consistency matrix (*Figure 1—figure supplement 2C*). When K is too high for the dataset, inconsistency will arise in seqNMF factorizations if $\lambda = 0$. This can be seen by a disruption in the consistency, which we tested by computing consistency matrices for K above the estimated K. (*Figure 1—figure supplement 2D*).

### Selecting a consistent factorization

For each dataset, we selected the most consistent factorization on which to perform all further analysis. Once we had selected an appropriate number of sequences for each dataset, using the analyses described above, we ran seqNMF 25 times at this value of $K$ from different random initial conditions, and picked the factorization that was most consistent with the other factorizations (*Figure 1—figure supplement 2D*). Factorizations at $K$ chosen by the above methods tended to be more consistent than factorizations at higher $K$ (*Figure 1—figure supplement 2D*).

### Significance testing for cross-correlation analyses

Several of our results involve analyzing the temporal relationship between different timecourses (factors and neurons; factors and song acoustic features; factor autocorrelations). These analyses involve testing the significance of the cross-correlation between two time series, compared to null cross-correlation values that could occur if the signals were circularly shifted relative to each other by a random large time lag. Before measuring cross-correlations, we centered each signal to have zero mean. If we are assessing the cross-correlation at lags in the range from −L to L, we want to compare values measured here to null values measured at random lags longer than L. We compute the cross-correlation at each lag $\ell$ in the range $-T < \ell < T$, where T is the length of the time series, by circularly

shifting one of the time series by $\ell$ and computing the dot product with the other time series. We then use the cross-correlations at null lags ($-(T-L) < \ell < -L$ or $L < \ell < (T-L)$) to determine a Bonferroni-corrected significance threshold. The threshold is the $100 \times (1 - p/Num)^{th}$ percentile of the absolute value of these null cross-correlations, where $Num$ is the number of comparisons (2 L times the number of tests being run), and $p$ is the $p$-value. Significance is achieved for lags at which the measured cross-correlation exceeds this value.

### Assessing song locking, the cross-correlation between each factor and acoustic song features

Several of our results involve quantifying the temporal relationship between sequence timecourses ($H$'s) and the song. To do this, we measured the cross-correlation of sequences with song acoustics using eight acoustic features common in the songbird literature (*Tchernichovski et al., 2000*): amplitude, entropy, pitch goodness, aperiodicity, mean frequency, pitch, frequency modulation, and amplitude modulation. Each of these acoustic features is measured from the song at 1ms resolution using standard software (Sound Analysis Pro, http://soundanalysispro.com/, *Tchernichovski et al., 2000*). The seqNMF H's are upsampled to this resolution, then cross-correlation between each $H$ and each song feature is assessed using the above procedure, with L=1 s, p=0.05, and Bonferroni correction (2000 timebins) × (8 features) × (K sequences). The overall measure of song locking is computed by integrating the number of seconds that a given sequence has a significant correlation with each of the song features.

### Assessing which neurons participate in each sequence

Several of our results involve assessing which neurons participate in each sequence. In order to do this, we measure whether there is a significant cross-correlation between each neuron and each factor (with L=0.5 s, p=0.05, and Bonferroni correction (30 timebins) × (N neurons) × (K sequences)). Note that, since seqNMF is run on the neural data, it is guaranteed that some neurons will be correlated with the factors —the primary aim of this test is to assess which neurons are in which sequences.

### Tracking HVC projection neurons over the course of major song changes

A core motivation for using calcium imaging methods instead of other methods was the possibility to track HVC projection neurons over the course of major song changes. HVC projection neurons are particularly difficult to record with electrophysiological methods—current methods are unable to record an HVC projection neuron for more than a few hours, and tend to record one, or at most three, projection neurons at a time (*Okubo et al., 2014*; *Okubo et al., 2015*). Previous studies of song-locked HVC activity throughout the learning process could only track changes in the neural population that occurred at a timescale slower than a week, because population statistics had to be compiled from single-neuron recordings (*Okubo et al., 2015*). This technique misses rapid changes that can happen within a day (*Tchernichovski et al., 2001*), and is unable to assess the stability of HVC sequences.

Stability of HVC sequences over time can be assessed using calcium imaging, though some challenges remain due to the potential for errors in tracking neurons across days. Single-photon calcium imaging methods have been used to address the stability of HVC sequences in adult birds with stable songs, observing stable song-locked activity in slightly more than half of HVC projection neurons, and unstable song-locked activity in slightly less than half of HVC projection neurons (*Liberti et al., 2016*). This measure is likely an underestimate of the stability of HVC activity, since noise in tracking cell locations across days could lead to perceived instability. Thus, HVC sequences appear relatively stable in birds with stable song, but what about birds whose songs are changing? The potential for errors in tracking neurons across days was one factor in our decision to record in birds undergoing very rapid learning. It was necessary for us to expand upon previous methods for tracking neurons recorded by calcium imaging over time (*Sheintuch et al., 2017*), likely due to the relatively short individual file sizes in our dataset from singing juvenile birds (we recorded many short files each day, when the birds happened to sing, instead of longer continuous files).

We tracked the activity of populations of HVC neurons over multiple days using Spatial Tracking Across Time (STAT, *Gu et al., 2023*). This method builds off of previous methods (*Sheintuch et al.,*

*2017*), where individual cell pairs' shape spatial correlation and distance are used to determine the correspondences between cells extracted from different sessions. STAT also considers local neighborhood motion consistency (*Belongie et al., 2002*; *Myronenko and Song, 2010*; *Ma et al., 2016*) in computing the optimal tracking of cells across sessions, and requires less manual supervision. Individual cell identity matchings are iteratively updated with respect to the local motion consistency cost function. Cells that have no good match are excluded, as are cells with the abnormal coefficient of variations (the standard deviation divided by the mean). The coefficient of variation exclusion criteria was designed to detect artifacts (blood vessels erroneously detected by the algorithm as a neuron, which has very different spatial and temporal profiles compared to actual neurons), and fewer than 10% of extracted cells were excluded. Finally, the results of the matching algorithm are checked manually.

### Tracking sequences extracted on one subset of a dataset to another subset of the dataset

In order to track a sequence, $W$, extracted in one subset of a dataset ($X_1$, for example before tutoring) to another subset of the dataset ($X_2$, for example after tutoring), we first mean-subtract $W$ and $X_2$ along the time dimension, then estimate $\widetilde{H}_2 = W^\top \circledast X_2$. In order to assess whether a neuron significantly participates in $W$ in dataset $X_2$, we bootstrap using control datasets $X_2^{shuff}$, in which data from each neuron is circularly shifted in time by a different random amount. We then ask whether the neuron participates more strongly in the real dataset compared to participation calculated on control datasets (p=0.05 significance threshold, Bonferroni corrected for the number of neurons and the number of time-lags). Specifically, we compare $\widetilde{W}_2 = X_2 \widetilde{H}_2^\top$ to $\widetilde{W}_2^{shuff} = X_2^{shuff} \widetilde{H}_2^{shuff\top}$.

### Assessing sequence coverage of song bouts

Sequence coverage quantifies the observation that sequences in isolated birds appear to pop on and off at somewhat arbitrary moments in bouts, leaving some sections of some bouts with no clear sequences present. First, the moments when each sequence occurs are estimated by computing when $\widetilde{H} = W^\top \circledast X$ is larger than expected by chance (Bonferroni-corrected 95% percentile of $\widetilde{H}^{shuff} = W^\top \circledast X^{shuff}$). Next, the sequence is convolved with the corresponding $W$. Finally, the total number of seconds when some sequences was present is divided by the total number of seconds in the bout, and multiplied by 100, to get the percent of the bout covered by some sequence. Note that sequence coverage is distinct from previously described measures of burst coverage within a repeatable adult song motif (*Lynch et al., 2016*).

## Acknowledgements

This work was supported by a grant from the Simons Collaboration for the Global Brain, the National Institutes of Health (NIH) [R01 DC009183], and the G Harold and Leila Y Mathers Charitable Foundation. ELM received support through the NDSEG Fellowship program and the Simons Society of Fellows. Special thanks to Andrew Bahle for his comments on earlier versions of the manuscript.

## Additional information

### Funding

| Funder | Grant reference number | Author |
| --- | --- | --- |
| National Institutes of Health | R01 DC009183 | Shijie Gu<br>Natalia I Denisenko<br>Michale S Fee |
| Simons Foundation | Simons Collaboration for the Global Brain | Shijie Gu<br>Emily L Mackevicius<br>Michale S Fee |
| G. Harold and Leila Y. Mathers Foundation | | Shijie Gu<br>Natalia I Denisenko<br>Michale S Fee |

| Funder | Grant reference number | Author |
|--------|------------------------|--------|
| Department of Defense | National Defense Science and Engineering Graduate Fellowship | Emily L Mackevicius |
| Simons Foundation | Society of Fellows | Emily L Mackevicius |
| Gatsby Charitable Foundation | GAT3708 | Emily L Mackevicius |

The funders had no role in study design, data collection and interpretation, or the decision to submit the work for publication.

## Author contributions

Emily L Mackevicius, Conceptualization, Data curation, Software, Formal analysis, Supervision, Investigation, Visualization, Methodology, Writing – original draft, Project administration, Writing – review and editing; Shijie Gu, Data curation, Software, Formal analysis, Validation, Visualization, Methodology, Writing – review and editing; Natalia I Denisenko, Investigation, Methodology; Michale S Fee, Conceptualization, Resources, Supervision, Funding acquisition, Methodology, Writing – original draft, Project administration, Writing – review and editing, Formal analysis

## Author ORCIDs

Emily L Mackevicius http://orcid.org/0000-0001-6593-4398
Shijie Gu http://orcid.org/0000-0001-6257-5756
Michale S Fee http://orcid.org/0000-0001-7539-1745

## Ethics

Animal care and experiments were carried out in accordance with NIH guidelines, and reviewed and approved by the Massachusetts Institute of Technology Committee on Animal Care (Protocol 0721-064-24: Chronic Recording of Neural Activity in Songbirds).

## Decision letter and Author response

Decision letter https://doi.org/10.7554/eLife.77262.sa1
Author response https://doi.org/10.7554/eLife.77262.sa2

# Additional files

## Supplementary files

• Transparent reporting form

## Data availability

Data and code to generate figures is publically available on the Dryad data-sharing platform, at the following URL: https://doi.org/10.5061/dryad.j0zpc86km.

The following dataset was generated:

| Author(s) | Year | Dataset title | Dataset URL | Database and Identifier |
|-----------|------|---------------|-------------|-------------------------|
| Mackevicius E, Gu S, Denissenko N, Fee MS | 2023 | Calcium imaging dataset from the pre-motor area HVC in singing zebra finches before and after tutor exposure | https://dx.doi.org/10.5061/dryad.j0zpc86km | Dryad Digital Repository, 10.5061/dryad.j0zpc86km |

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
