## [Editor Report]

This fundamental work in songbirds shows that stereotyped neural sequences known to drive the correspondingly stereotyped acoustic structures of adult songs can exist very early in development even when songs are variable and before birds have been provided song models by tutors. The evidence is exceptional and includes imaging activity of populations of premotor neurons in singing birds. This paper provides important insights into the mechanistic foundations of how nature and nurture work together to produce learned motor sequences.

---

## [Decision Letter]

**Decision letter after peer review:**

Thank you for submitting your article "Self-organization of songbird neural sequences during social isolation" for consideration by *eLife*. Your article has been reviewed by 3 peer reviewers, one of whom is a member of our Board of Reviewing Editors, and the evaluation has been overseen by Ronald Calabrese as the Senior Editor. The following individual involved in the review of your submission has agreed to reveal their identity: Yarden Cohen (Reviewer #2).

Essential revisions:

1) In the abstract and elsewhere, the authors write that 'experience is not necessary for the formation of sequences.' This statement is a bit of a reach – this paper exclusively shows that tutor exposure is not necessary for chain formation. It's conceivable that the experiences of deafening, muting or the absence of singing WOULD block chain formation. It's also possible that cohabitation with females – who call – could play some rudimentary auditory experience that helps establish chains. Please edit the language of the manuscript to make sure the conclusions match specific experimental results. Comb the manuscript for use of the work 'experience' and ensure that tutor-experience is what's written.

2) The adequacy of the sequence detection method (seqNMF) and analyses of its outcomes need further explanation and support. This is especially needed when describing results where sequences are truncated, jittery, or otherwise variable (as some of the results indicate). The presentation of results will be strengthened by:

2.1. A clear presentation of seqNMF's outcomes and fit to data:

2.1.1. Explaining in the main text and methods what is meant by 'sequences' that the algorithm extracts. It is not clear if these are cells activating one after the other or any robust spatiotemporal pattern. seqNMF allows seeking 'event based' or 'part based' factorization. Please describe which was used in this manuscript.

2.1.2. How much of the data variability is explained by sequences?

2.1.3. How specific are neurons' activity to sequences (compared to its activity not in sequences).

2.2. Control analyses (or citation if shown elsewhere) can show that the atypical properties of sequences are not confounded by seqNMF.

2.2.1. For example, measures in Figure 1E-K may be compared to sequences extracted from time-shuffled data. (Similar to the 'sequenciness' approach defined by previous work of the authors[3]).

2.2.2. Alternatively, if at all possible (because data is limited), results could be compared to analyses carried out on held-out data. For example, sequences can be discovered in training set data and used to calculate results as in Figure 1E-K on test set data.

2.3. Is it possible to compare sequences (the W's) found before and after training? The claim that they persist needs quantitative support.

3) The tutoring process and its effects need a clearer presentation.

3.1. The methods are vague about the process of tutoring (specifically, how many days of tutoring each bird received).

3.2. When describing (in text and in figure panels) the effect of tutoring it is most helpful to show: (3.2.1) the tutors template, (3.2.2) parts of the template that were copied by the tutee. Currently, the manuscript shows newly adopted syllables but doesn't demonstrate that these syllables were copied from the tutor. (3.2.3) The imitation score. These elaborations on tutor match can be put in a supplemental figure.

4) In all figures the spectrograms are tiny and it is very difficult to see the link between the identified sequences and the acoustic structure of song. Please revise figures so that the reader does not have to simply depend on somewhat abstracted statistical measures of song locking to absorb the result. Please make spectrograms bigger in the figures.

Also In Figure 1C, the first and second sequences seem to overlap. E.g., for the second sequence (magenta) the units in the upper sequence also participate. I assume that whether the seqNMF algorithm generates these two sequences or merges them is dependent on the parameters? But either way, how do we interpret these sequences: is the conclusion that the same units participate in more than one sequence (in the same order) or that it is the same, but noisy sequence? How often were different there shared sub-sequences between the identified sequences?

---

## [Author Response]

Essential revisions:1) In the abstract and elsewhere, the authors write that 'experience is not necessary for the formation of sequences.' This statement is a bit of a reach – this paper exclusively shows that tutor exposure is not necessary for chain formation. It's conceivable that the experiences of deafening, muting or the absence of singing WOULD block chain formation. It's also possible that cohabitation with females – who call – could play some rudimentary auditory experience that helps establish chains. Please edit the language of the manuscript to make sure the conclusions match specific experimental results. Comb the manuscript for use of the work 'experience' and ensure that tutor-experience is what's written.

The reviewer makes an important point. Our results apply to tutor experience specifically, and do not speak to the possible effects of other types of experience on HVC chain formation. We have edited the abstract to use “tutor experience” instead of “experience” when referring to our work. We combed through the manuscript for cases where “experience” is discussed, and changed this to “tutor experience” wherever the discussion referred to our results.

2) The adequacy of the sequence detection method (seqNMF) and analyses of its outcomes need further explanation and support. This is especially needed when describing results where sequences are truncated, jittery, or otherwise variable (as some of the results indicate). The presentation of results will be strengthened by:2.1. A clear presentation of seqNMF's outcomes and fit to data:

We apologize for the lack of clarity, and have expanded our discussion of the seqNMF method, as detailed below.

2.1.1. Explaining in the main text and methods what is meant by 'sequences' that the algorithm extracts. It is not clear if these are cells activating one after the other or any robust spatiotemporal pattern. seqNMF allows seeking 'event based' or 'part based' factorization. Please describe which was used in this manuscript.

We added the following paragraph to the main text to provide additional information about the seqNMF algorithm, and what is meant by ‘sequences’:

The seqNMF algorithm is designed to find patterns in data, and identify the times when each pattern occurs. It is a generalization of non-negative matrix factorization, allowing for non-synchronous patterns of neural activity, for example sequential firing of neurons in a chain. In the context of seqNMF, a sequence is defined as a pattern of activity in a population of neurons that extends for more than one time step. The algorithm allows for an individual neuron to be active at more than one time in a sequence, but in practice neurons in HVC or the hippocampus are typically active only once in a sequence. Each seqNMF factor is represented as an exemplar sequence ***W***, and a vector of times when the sequence occurs, ***H***. A sequence in the data is factorized as the convolution of a ***W*** with its corresponding ***H***. Data often contain multiple sequences, each represented by different ***W***s and ***H***s, and the full data matrix is represented as a sum across all different extracted sequences. Using gradient descent, the algorithm automatically finds sequences that best fit the data, while avoiding redundancies between sequences. It allows for explicitly selecting between ‘parts-based’ and ‘events-based’ factorizations, but here we are agnostic to this distinction, selecting the factorization that best fits the data. SeqNMF has been shown to be robust to several forms of neural variability MackeviciusBahle 2018. In addition, it includes a measure of “sequenciness”, which quantifies the extent to which data are better explained by temporally extended sequences than by synchronized activity. This measure ranges from zero (only synchronized activity) to one (only temporally extended sequences, such that shuffling timebins in the data erases all repeatable structure).

In addition, we added this clarification to the methods:

It is possible for seqNMF to extract non-sequential (purely synchronous) patterns of neural activity. SeqNMF measures the extent to which a dataset contains sequential vs. synchronous patterns using a “sequenciness” score, which ranges between 0 and 1. A score of zero indicates that all explanatory power of the factorization is due to synchronous activity, while a score of one indicates that none of the explanatory power is due to synchronous activity. Quantitatively this is calculated by shuffling the timebins in the dataset. In these datasets, both “sequenciness” scores and visual inspection of the factors suggest the presence of temporally extended patterns, which we refer to as sequences.

2.1.2. How much of the data variability is explained by sequences?

On average, 45.21%+-7% of neuronal activity was specific to sequences. We calculated this by binarizing the seqNMF reconstruction and binarizing the data, and computing the correlation between these two matrices, then dividing the norm of the correlation by the norm of the data. As in other analyses, W and H are thresholded by 25% of the 99th percentile. We have added this information to the text.

2.1.3. How specific are neurons' activity to sequences (compared to its activity not in sequences).

In order to show that atypical sequences in isolate birds are not confounded by seqNMF, control analyses were performed by constructing time-shuffled datasets, as in the reviewer’s suggestion 2.2.1, see detail below.

2.2. Control analyses (or citation if shown elsewhere) can show that the atypical properties of sequences are not confounded by seqNMF.

In order to show that atypical sequences in isolate birds are not confounded by seqNMF, control analyses were performed by constructing time-shuffled datasets, as in the reviewer’s suggestion 2.2.1, see detail below.

2.2.1. For example, measures in Figure 1E-K may be compared to sequences extracted from time-shuffled data. (Similar to the 'sequenciness' approach defined by previous work of the authors[3]).

We repeated each of the analyses in Figure 1 for time-shuffled data (independent circular shifting of each neuron, as in the “sequenciness” approach). In the summary plots (participating neurons, reliability, song locking, and coverage), we now include a red line indicating the median value obtained on shuffled data. The results of these control analyses are consistent with the view that seqNMF detects sequences in isolated birds substantially more than expected by chance (time-shuffled data), but these sequences appear less robust than in the adult bird.

2.2.2. Alternatively, if at all possible (because data is limited), results could be compared to analyses carried out on held-out data. For example, sequences can be discovered in training set data and used to calculate results as in Figure 1E-K on test set data.

Given the limited amount of data in pre-tutoring birds, we prefer the control analyses suggested by the reviewer in 2.2.1.

2.3. Is it possible to compare sequences (the W's) found before and after training? The claim that they persist needs quantitative support.

We quantified the correlation between the W’s found before and after tutoring, and are including this information in the main text. We performed two control analyses. In one analysis, we shuffled the identity of the neurons within Wpre. In the second analysis, we circularly shifted the time dimension of each of the neurons within Wpre. In all four learner birds, the correlation between the pre-tutoring and post-tutoring sequences was significantly higher than expected by a time-shuffled control (p=0.014, p=0.018, p=0.0004, p=0.0002, respectively). Additionally, the correlation between pre- and post- tutoring sequences was significantly higher than a neuron-shuffled control in two birds (p=0.0099, p=0.0032), and trended higher in the remaining two birds (p=0.077, p=0.079). We believe the weaker effect in these two birds is because many neurons fire at similar (synchronous) latencies in these birds, one of which is shown in Figure 4c. We have included this information in the text.

3) The tutoring process and its effects need a clearer presentation.3.1. The methods are vague about the process of tutoring (specifically, how many days of tutoring each bird received).

Each bird received at least 7 sequential days of tutoring, and the data shown in the paper were acquired during this period. We have clarified this information in the text.

3.2. When describing (in text and in figure panels) the effect of tutoring it is most helpful to show: (3.2.1) the tutors template, (3.2.2) parts of the template that were copied by the tutee. Currently, the manuscript shows newly adopted syllables but doesn't demonstrate that these syllables were copied from the tutor. (3.2.3) The imitation score. These elaborations on tutor match can be put in a supplemental figure.

We thank the reviewers for this suggestion, and have added new supplemental figures for each of the learner birds showing the tutor template, the imitation score, and which syllables in the tutor song are imitated. In each case, the new syllables that emerge after tutoring appear to be imitating syllables in the tutor song, and appear to persist, sometimes differentiating into new variants, and ultimately be used in the learned song.

4) In all figures the spectrograms are tiny and it is very difficult to see the link between the identified sequences and the acoustic structure of song. Please revise figures so that the reader does not have to simply depend on somewhat abstracted statistical measures of song locking to absorb the result. Please make spectrograms bigger in the figures.

We have made the spectrograms larger in all of the figures. In addition, we have included supplemental figures showing large spectrograms and corresponding neural data for each bird for a 6-second bout of pre-tutoring singing data.

Also In Figure 1C, the first and second sequences seem to overlap. E.g., for the second sequence (magenta) the units in the upper sequence also participate. I assume that whether the seqNMF algorithm generates these two sequences or merges them is dependent on the parameters? But either way, how do we interpret these sequences: is the conclusion that the same units participate in more than one sequence (in the same order) or that it is the same, but noisy sequence? How often were different there shared sub-sequences between the identified sequences?

In this bird, the algorithm factorizes this feature of the data as two distinct sequences that are sometimes produced at different times, and sometimes produced concurrently. The possibility of overlapping subsequences is very interesting, and is reminiscent of splitting sequences observed in Okubo et al. It’s clear that this happened in this isolate bird, but examining the other birds, it doesn’t seem to happen in other birds (see new supplemental data figures, showing enlarged spectrograms and neural data for each bird).